# The Current Status of Indigenous Ovine Genetic Resources in Southern Africa and Future Sustainable Utilisation to Improve Livelihoods

**Annelin Henriehetta Molotsi [1,]*[] , Bekezela Dube [2] and Schalk Willem Petrus Cloete [1,3]**

[1]  Department of Animal Sciences, Stellenbosch University, Stellenbosch, Private Bag X1, Stellenbosch 7600, South Africa; schalkc2@sun.ac.za

[2]  Animal Production, Agricultural Research Council, Private Bag X2, Irene 0062, South Africa; dubeb@arc.agric.za

[3]  Directorate Animal Sciences, Western Cape Department of Agriculture: Elsenburg, Private Bag X1, Elsenburg 7607, South Africa

*  Correspondence: annelind@sun.ac.za; Tel.: +27-218-083-148

**Abstract:** Indigenous sheep play an important role in the livelihoods of communal farmers in the Southern Africa Development Community (SADC), and this underlines the need to curb the genetic erosion of these valuable resources. This contribution reports that the phenotypic performance and genetics gains of institutional and commercial sheep in Southern Africa are well recorded. In contrast, there is a dearth of knowledge as far as the performance and genetic gains of indigenous ovine genetic resources utilized by smallholder farmers are concerned. High levels of genetic diversity have been observed in exotic breeds, whereas low levels of genetic diversity were found in the Zulu and Namaqua Afrikaner breeds. Phenotypic measurements for indigenous resources include linear measurements indicative of size and reproduction for Zulu sheep. Lamb survival, reproduction and resistance to ticks of the indigenous, fat-tailed Namaqua Afrikaner sheep, as well as growth and reproduction have also been recorded for Sabi and Landim sheep. This review discusses ways to sustainably utilize ovine genetic resources, which includes the suggested implementation of structured breeding and conservation programs, marketing, improving feed resources, health and diseases, as well as gender and age issues. Clearly, there is ample room for further research and development as far as the performance and improvement of African indigenous sheep are concerned.

**Keywords:** breeding; conservation; genetic diversity; genetic erosion; robustness; socio-ecological systems

## 1. Introduction

According to international standards, the bulk of the agricultural landscape in Southern Africa is arid or semi-arid. Being able to adapt to arid environments, sheep markedly contribute to the livelihood of farmers in Southern Africa in these regions. The Southern Africa Development Community (SADC) consists of Angola, Botswana, Comoros, Congo, Eswatini, Lesotho, Madagascar, Malawi, Mauritius, Mozambique, Namibia, Seychelles, South Africa, Tanzania, and Zimbabwe [1]. In the SADC region, there are 39 million sheep, of which 19.9 million are found in South Africa [2]. There are 109 different sheep breeds in Southern Africa, where South Africa has the highest number (46) against other SADC countries [3]. The ovine genetic resource is genetically diverse, including specialist wool and meat breeds, dual-purpose breeds, terminal sire breeds, and adapted indigenous fat-tailed breeds. Proven genetic progress has been attained in research and industry flocks for economically important growth, reproduction, and fibre traits [4]. However, slow or even no genetic progress has been reported for the

indigenous ovine genetic resources used mainly in smallholder flocks. The major indigenous sheep breeds have experienced genetic erosion due to the interest of humans to select for production of wool and/or meat. Genetic erosion occurs when a population is subjected to genetic drift and inbreeding or is diluted by injudicious crossbreeding with exotics [1]. Such practices result in a loss of genetic variation and a decrease in fitness [5]. Almeida [6] emphasized that improvement of indigenous breeds through the importation of exotic breeds, especially from Europe, as well as the development of composite breeds, may have contributed to the genetic erosion of indigenous breeds. The problem with genetic erosion of livestock species is that it leads to an inability of the populations in question to adapt to stressful environmental conditions. The effects of climate change are tangible in the SADC region. Having livestock that are able to adapt to these environments is, therefore, crucial. This is specifically true in regions of the SADC, where trypanosomiasis caused by the tsetse fly is prevalent. Increases in temperature have led to a decrease in tsetse flies in the Zambezi valley of Zimbabwe [7], causing the tsetse flies to relocate to the higher cooler regions of Zimbabwe. This is also true for other tick-borne and internal parasite diseases in sheep [8] as well as with rift valley fever transmitted by mosquitoes [9]. Increases in temperature can lead to heat stress of animals and negatively influence their growth, reproduction, and survival [10]. Since communal farmers in the SADC region are dependent on livestock for providing social and economic security, there is a need to ensure that these ovine genetic resources remain robust and are sustained. Robustness is defined as the ability of an animal to adapt to challenging environments [11] and encompasses traits such as reproduction, survival, parasite and disease resistance, heat and drought tolerance [12].

This review, therefore, focuses on indigenous ovine genetic resources in Southern Africa (the SADC region), the progress made and the way forward for sustainable usage of these resources.

## 2. Ovine Genetic Resources in SADC

The landscape of Southern Africa can be divided into wet humid conditions on the eastern coast and the drier northern regions (Figure 1). Breeds found in the wet, eastern coast are the thin and fat-tailed, smaller framed Nguni sheep breeds comprised of the Zulu, Swazi, Pedi, and Landim (Mozambique) sheep. In the Northern drier SADC regions fat-tailed and fat-rumped sheep breeds are found, and these are the Damara (Namibia and Northern Cape), Namaqua Afrikaner, Ronderib Afrikaner, and Sabi (Zimbabwe). Table 1 depicts all the major indigenous, synthetic, and developed breeds prevalent in the SADC regions. Synthetic breeds refer to breeds that have been founded through crossbreeding programs; whereas, developed breeds refer to exotic breeds imported to the SADC region and that have been selected for certain traits desired in SADC environmental conditions. The population numbers for the exotic transboundary breeds are well recorded in South Africa, Botswana, Namibia, and Tanzania. However, population sizes of indigenous breeds in most of the other SADC countries are commonly unknown, except in Tanzania, where a census is routinely done at the species level [13]. The population numbers in Table 1 obtained from the Food and Agricultural Organisation of the United Nations—Domestic Animal Diversity Information System (FAO-DAD-IS) database [13] has been updated between the period of Feb 1990 to Feb 2018. The effective population sizes of some indigenous breeds are known, such as the Namaqua Afrikaner (15 breeding males and 300 breeding females updated in 2006); Tswana sheep from Botswana (6500 breeding males and 13,000 breeding females recorded in 1990); Sabi sheep (30 breeding males and 270 breeding females recorded in 2018 in Zimbabwe) while it remains unknown for the majority of indigenous breeds. Three sheep breeds in SADC are already extinct, the Hottentot, White Wooled Mountain, and the Wooled Persian; whereas, the Pedi, Van Rooy, Gellaper, and Veldmaster are at risk of endangerment. To ensure the survival of indigenous ovine resources in SADC, it is important that the effective population size of these breeds are maintained and recorded. Here the FAO-DAD-IS database can play an integral role in storing data on indigenous breeds. Various role players in government and research institutions can play a role in ensuring that the database remains updated. The use of genomics to further characterize indigenous ovine resources is also valuable to ensure sustainable utilization [1]. Various population structure

studies using genomics have been done to determine the genetic diversity between and within local breeds [14–16]. Genome-wide association studies and functional genomic studies have also been conducted on the Damara breed to identify genes associated with polyceraty [17]. Sheep breeds in the SADC region have been phenotypically described, and some characteristics are known on their growth and reproduction performance; however, due to limited recordkeeping by farmers keeping these breeds, little information is available on the additive genetic variation observed for important traits. Indigenous breeds that are kept at research stations have been recorded and measured for growth and production traits. These research stations include Matopos in Zimbabwe (Sabi), Chobela research station in Mozambique (Landim), Carnarvon research station in Northern Cape of South Africa (Namaqua Afrikaner), and the research farms and communities around the University of Zululand (Zulu). A detailed outline of the phenotypic description of the Tswana, Swazi, Sabi, Landim, and Black headed Persian breeds has been provided by Wilson [18]. The availability of information on the genetic progress of indigenous ovine resources is necessary to ensure sustained utilization of these resources by future generations. Therefore, the next subsection of this paper will focus on the genetic diversity of ovine genetic resources in the SADC, their phenotypic characterization, and traits important for selection.

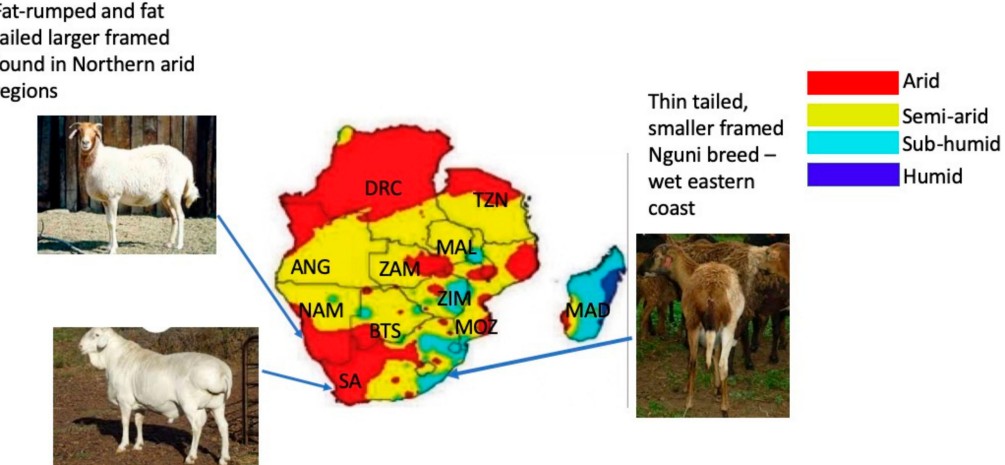

**Figure 1.** Map depicting the climatic regions of the Southern Africa Development Community (SADC)region as well as the presence of ovine genetic resources according to typical distribution. *ANG -Angola; BTS—Botswana; DRC—Democratic Republic of Congo; MAD—Madagascar; MALMalawi; MOZ—Mozambique; NAM—Namibia; SA—South Africa; TZN—Tanzania; ZAM—Zambia.

**Table 1.** List of indigenous sheep breeds present in the different Southern African Development Communities (SADC) countries and their estimated risk level. Adapted from the FAO-DAD-IS database [13].

| Breeds | SADC Country | Exotic/Indigenous/Synthethic | Risk Level | Population Numbers Minimum (Updated between 1990–2018) | Population Numbers Maximum (Updated between 1990–2018) |
|---|---|---|---|---|---|
| Africaner | South Africa | Indigenous | Unknown | Unknown | Unknown |
| Afrino | South Africa | Synthetic (South African Mutton Merino 50%, Blinkhaar Afrikaner 25%, Merino 25%) | Unknown | 1000 | 3141 |
| Angola long-legged | Angola | Indigenous | Unknown | Unknown | Unknown |
| Angola maned | Angola | Indigenous | Unknown | Unknown | Unknown |
| Bahu | Democratic republic of Congo | Indigenous | Unknown | Unknown | Unknown |
| Baluba | Democratic republic of Congo | Indigenous | Unknown | Unknown | Unknown |
| Basotho sheep | Lesotho | Indigenous | Unknown | Unknown | Unknown |
| Bezuidenhout | South Africa | Synthetic/Indigenous (derived from Ronderib Afrikaner and wooled Persian in 1917) | Unknown | 20 | 100 |
| Blackheaded Persian | Angola, South Africa, Zimbabwe, Botswana, Namibia, Mozambique, Tanzania, Mauritius | Indigenous (transboundary breed) | Not at risk | 12,100 | Unknown |
| Blinkhaar Ronderib Afrikaner | South Africa, Namibia | Indigenous | Unknown | 1000 | Unknown |
| Boesmanlander | South Africa | Developed breed | Unknown | Unknown | Unknown |
| Damara | South Africa, Botswana, Namibia, | Indigenous (transboundary breed) | Not at risk | 6100 | 20,114 |
| Dormer | South Africa | Synthetic (transboundary breed) | Not at risk | 1000 | 35,707 |
| Dorper | South Africa, Zimbabwe, Malawi, Botswana, Namibia, Angola, Swaziland and Mauritius | Synthetic/Indigenous (transboundary) | Not at risk | 606,100 | 1304,810 |
| Dohne Merino | South Africa, Namibia | Synthetic (transboundary) | Not at risk | 1000 | 316,454 |
| Hottentot | South Africa | Indigenous | Extinct | 0 | 0 |
| Karakul | South Africa, Botswana, Namibia, Mozambique, Angola, | Exotic (transboundary breed) | Not at risk | 160,120 | 240,785 |
| Meatmaster | South Africa, Namibia | Synthetic | At risk | 3000 | 15,927 |
| Man'gati | Tanzania | Indigenous | Not at risk | 5000 | 10,000 |
| Merino | South Africa, Malawi, Lesotho, Angola | Exotic | Unknown | 15,000,000 | Unknown |
| Mondombes | Angola | Indigenous | Unknown | Unknown | Unknown |
| Merino Landsheep | South Africa | | Not at risk | 1000 | 25,294 |
| Mouton indigène moyen à courte patte | Democratic republic of Congo | Indigenous | Unknown | Unknown | Unknown |

**Table 1.** *Cont.*

| Breeds | SADC Country | Exotic/Indigenous/Synthethic | Risk Level | Population Numbers Minimum (Updated between 1990–2018) | Population Numbers Maximum (Updated between 1990–2018) |
|---|---|---|---|---|---|
| Mouton indigène à longue patte | Democratic republic of Congo | Indigenous | Unknown | Unknown | Unknown |
| Namaqua Afrikaner | South Africa | Indigenous | Unknown | 100 | 1000 |
| Nguni | South Africa, Malawi, Mozambique, Swaziland | Indigenous | Unknown | 109,800 | Unknown |
| Pedi | South Africa | Indigenous | At risk | 120 | 342 |
| Ronderib Afrikaner | South Africa | Indigenous | Unknown | 10,000 | Unknown |
| Ronderib Merino | South Africa | indigenous | Unknown | Unknown | Unknown |
| South African Mutton Merino | South Africa, Zimbabwe, Namibia | Developed in SA (transboundary breed) | Not at risk | 1000 | 258,647 |
| Steekhaar | South Africa | Indigenous | Unknown | 100 | Unknown |
| Van Rooy | South Africa, Zimbabwe, Namibia | Synthetic—(Ronderib Africander and Rambouillet) (transboundary) | At risk | 1000 | 1081 |
| Vandor | South Africa | Synthetic—(Rambouillet and Van Rooy) | Unknown | 100 | Unknown |
| Walrich Vleismerino | South Africa | Developed in SA | Extinct | 0 | 0 |
| White Dorper | South Africa | Synthetic | Unknown | Unknown | Unknown |
| White wooled Mountain | South Africa | Developed in SA | Extinct | 0 | 0 |
| Wooled Persian | South Africa | Origin from Arabia | Extinct | 0 | 0 |
| Sabi | Zimbabwe | Indigenous (transboundary) | At risk locally | 50 | 300 |
| Tswana | Zimbabwe, Botswana | Indigenous (transboundary) | Not at risk | 250,000 in Zim | Unknown |
| Wiltiper | Zimbabwe | Synthetic (Wiltshire horn and blackhead Persian) | Unknown | 12,000 | Unknown |
| Primitif | Comoros | Indigenous | Unknown | Unknown | Unknown |
| Gellaper | Namibia | Indigenous | At risk | 100 | 2000 |
| Veldmaster | Namibia | Indigenous | At risk | 2000 | 6000 |
| Landim | Mozambique | Indigenous | Unknown | 150,000 | Unknown |
| East African Blackheaded | Tanzania | Indigenous | Not at risk | 800,000 | 1979,952 |
| Gogo Tanzania long-tailed | Tanzania | Indigenous | Not at risk | 3000,000 | 5182,627 |
| Kasulu | Tanzania | Indigenous | Not at risk | 200,000 | 300,000 |
| Maasai | Tanzania | Indigenous | Not at risk | 600,000 | 1500,000 |
| Rodrigaise | Mauritius | Indigenous | Unknown | 780 | Unknown |
| Sukuma | Tanzania | Indigenous (transboundary) | Not at risk | 1000,000 | 1500,000 |
| Tanzania Long-tailed | Tanzania | Indigenous (transboundary) | Not at risk | 2000 | 3000 |
| Urambo | Tanzania | Indigenous | Not at risk | 100,000 | 200,000 |
| Zunu | Angola | Indigenous | Unknown | Unknown | Unknown |

## 2.1. Genetic Diversity of Ovine Genetic Resources in SADC

The genetic diversity of 21 local sheep breeds (the Blackhead Persian, Blackhead Speckled Persian, Redhead Persian, Redhead Speckled Persian, Karakul, Damara, Namaqua Afrikaner, Ronderib Afrikaner, Pedi, Swazi, Zulu, Van Rooy, Dorper, Afrino, Dormer, Dohne Merino, Letelle, South African (SA) Mutton Merino, SA Landsheep, and SA Merino), as based on microsatellites, was studied by Soma et al. [14]. This study grouped the breeds into three categories, fat rumped (Persian and Van Rooy), fat-tailed (Damara, Afrikaner, Karakul, Zulu, Swazi, and Pedi), and wool types (Merinos). Persians exhibited low levels of genetic diversity (heterozygosity ranging from 0.401 to 0.520) due to their small founder population numbers in South Africa, as they originated from Somalia. Fat-tailed breeds also exhibited low levels of genetic diversity (0.480 to 0.637). The low heterozygosity of fat-tailed sheep was confirmed by Sandenbergh et al. [16] for the Damara and Namaqua Afrikaner by using single nucleotide polymorphic (SNP) markers. The wool type breeds of European origin exhibited the highest genetic diversity levels (0.527 to 0.711). The high levels of genetic diversity for Merino breeds have been confirmed by molecular studies using SNP markers [15,16]. Zulu sheep have been characterised both phenotypically [19] and genetically [19,20]. Concerns pertinent to the Zulu sheep included possible genetic erosion due to crossbreeding with Dorpers, Damaras, and Merinos. However, a recent study by Selepe et al. [20] indicated high levels of inbreeding in four populations of Zulu sheep sampled in the KwaZulu Natal province and that these populations were genetically isolated. The Zulu sheep kept on two research stations exhibited high genetic diversity and low levels of inbreeding. Introgression with the Merino, Dorper, and Damara was observed by Selepe et al. [20]. Therefore, the only concern would be to increase the effective population size of the Zulu breed and to implement strategies to enable this. Dorpers kept by smallholder farmers in the Western Cape region indicated some introgression with fat-tailed breeds (represented by the Namaqua Afrikaner) and wool breeds (represented by the SA Mutton Merino) [21].

## 2.2. Phenotypic Characterisation and Traits Important for Selection

Local and international ovine productivity hinges on several traits within the economically important growth, reproduction, fibre, and disease resistance trait complexes. South African institutional and commercial genetic resources were assessed both for phenotypic performance, as well as genetic gains, in most trait complexes (growth, fibre traits and reproduction; see review by Schoeman et al. [4]). These authors reported marked genetic gains in the genetic merit of the top-performing breeders in all commercial breeds. Studies on institutional resource flocks also found genetic variation in resistance to stressors, including gastro-intestinal parasites and breech flystrike (see review by Cloete et al. [22]). Even though selection to improve production in exotic breeds has proven to be successful, it has also led to genetic erosion of the diversity within purebred ovine genetic resources [23]. This is often associated with inbreeding depression, which results in a decrease in the robustness of animals. However, the situation in indigenous ovine genetic resources cared for by smallholder farmers is less well recorded and thus summarised below.

The Zulu breed has been assessed for body measurements, such as wither height, heart girth, live weight, and scrotal circumference [24]. The first three measurements are a good indication of growth, whereas scrotal circumference indicates ram fertility and may be an indicator of its daughters' reproductive performance [25]. In cattle, scrotal circumference is favourably correlated to the age at first calving [26]. The mature live weight for Zulu ewes and rams is 32 and 38 kg, respectively, which is small compared to the commercial breeds. Kunene [24], however, contended that Zulu sheep adapt in hot and humid climates, while also exhibiting good disease resistance. These characteristics of the Zulu breed have not been tested but were observed by those smallholder farmers interviewed. Information regarding the reproductive ability of Zulu sheep is lacking.

Reproduction of the Namaqua Afrikaner has been quantified and compared to other breeds. The Namaqua Afrikaners achieved higher survival from birth to weaning (91%) in comparison to the Dorper (88%) [27]. Namaqua Afrikaner ewes also outperformed Dorpers and South African

Mutton Merinos for the number of lambs weaned per ewe lambed, but not for the weight of lamb weaned per ewe lambed [28]. In addition, the Namaqua Afrikaner sheep were more resistant to ticks than the commercial breeds in the latter study. However, the commercial breeds outperformed the indigenous breeds for carcass yield and composition [28,29]. The Damara and Dorper showed higher conception rate and weaning percentage in relation to the Australian Merino at eight months of age. However, at 33 months, the Merino outperformed the Damara and Dorper for weaning percentage [30]. The reason for the difference in conception rate was due to the fat accumulation in the tails of the Damara, which made it more difficult for the rams to serve these ewes. Despite the unfavourable carcass yield for indigenous breeds, the Damara had higher concentrations of polyunsaturated fatty acids in the longissimus dorsi in comparison to the Dorper and Merino [31]. A number of functional genomics studies have indicated that the Damara does have an unique fatty acid metabolism [32]. In comparison to the Barbados blackbelly sheep in Malaysia, the Damara exhibited lower gastro-intestinal infection [33]. Tolerance to seasonal weight loss, which is an important adaptation trait during seasonal droughts, was exhibited by the Damara and Dorper breeds in relation to the Australian Merino [34,35].

Genetic parameters have been estimated for growth, reproduction, and survival traits of Sabi sheep [36]. The reproduction traits studied included fertility, which was a measure of whether the ewe lambed or not (88%), and prolificacy, which was measured by the number of lambs born per ewe lambing (1.2) [36]. On the other hand, Landim sheep were found to have a higher litter size of 1.38 lambs in comparison to the black headed Persian, which had a litter size of 1.0 lamb per parturition [37].

It is, thus, evident that further studies on the phenotypic assessment of indigenous African ovine genetic resources are urgently needed. There is, however, strong evidence that indigenous breeds exhibit robustness and could be utilized in farming systems geared to exploit these traits. Any intervention that does not consider this reality is unlikely to be successful in aiding sustainable development in this sector.

## 3. Sustainable Utilisation of Ovine Genetic Resources

Since indigenous ovine resources can play a pivotol role in the livelihood of communal farmers, their utilization should not be approached in isolation to the social-ecological system within which they are found. A social-ecological system refers to the interaction between the social and ecological spheres. It can be debated that the reason why little genetic improvement has been made with indigenous ovine genetic resources is that the interaction between the social and ecological spheres was not explored and that only one-sided solutions were brought to the table. Some of these factors that hinder viable utilization of these ovine genetic resources in rural communities include poor feed resources, poor record keeping, lack of structured breeding programs, disease and health challenges, poor market access, ageing male farmers since the youth and females are unwilling to farm, as well as a lack of technical skills and knowledge. Smallholder farmers practice communal farming, or those benefiting from the land reform and redistribution program (LRAD) farm with the assistance of a mentor [38]. The mentor is a commercial farmer who already has a successful farming enterprise. The issues discussed below can be managed best if farmers are grouped into cooperatives so that resources can be allocated efficiently. This intervention will allow sourcing of services and funds from government and other agencies. However, farmers should first comprehend the benefit of participating in such cooperatives. Engaging farmers through incorporating indigenous knowledge systems could ensure their involvement and facilitate the adoption of ideas to improve productivity and, hence, sustainable utilization of indigenous ovine resources.

### 3.1. Implementation of Structured Breeding Programs

Characterisation of indigenous breeds has been widely done in SADC regions under the African Union—Interafrican Bureau for Animal Resources (AU-IBAR) project. However, constantly monitoring the population sizes of these breeds remains a challenge. The FAO has developed the domestic animal diversity information system (DAD-IS) database where information about all the animal genetic

resources globally can be recorded and updated. This database is lacking information on the role and cultural value of breeds, their performance records, as well as fairly accurate population sizes of breeds in SADC countries [39]. The biggest challenge facing structured breeding in the smallholder sheep sector is the routine recording of phenotypic information within the constraints imposed by limited resources. One way to address this issue is by the establishment of community-based breeding programs, as in Ethiopia [40]. This intervention requires the need for people in the field that can assist farmers to design tailored breeding programs for their specific smallholder farms. This will allow farmers to create linkage among flocks by exchanging genetic material. To encourage recordkeeping, a recordkeeping application should be developed. Firstly, farmers need to be alerted to the value unlocked by the records they obtain from their animals every year by assisting them to monitor and manage their flocks. Workshops, where interactions with farmers will develop trust among the stakeholders and result in the adoption of the suggested practices, will need to be provided. Then, training programs can be arranged, starting with recordkeeping, followed by other aspects involving production. Assistance should be provided in the form of animal identification techniques, as farmers often use obsolete and ineffective methods to identify animals. These can be in the form of ear tags or tattooing pliers provided to the farmers.

### 3.2. Conservation of Genetic Resources

A secondary objective would be the conservation of the scarce and potentially valuable genetic resources [24]. Conservation of indigenous breeds should be based on both the genetic diversity and utility of the breeds. The latter criteria will include the threat status of the breed(s) and well as their contribution to the livelihood of farmers [41]. Based on the population admixture observed for Nguni breeds, Selepe, et al. [20] contended that conserving one or a few institutional populations from research stations will not be sufficient to represent the wide genetic diversity resident in local indigenous sheep. To enable this, analysts should identify and use flocks that contribute more to the observed genetic diversity. On the other hand, the social-economic value of the breed needs to be considered when choosing whether to conserve the breed or not. The low levels of genetic diversity in the Namaqua Afrikaner population [15,42] could be attributed to the small population size and geographic distribution of conservation flocks across the country. Therefore, genetic material from this breed should flow from the current conserved populations to smallholder farmers. Many farmers use large-frame exotic rams as terminal sires to increase the carcass value of the progeny of indigenous dams [43]. Such farmers typically share common water points and pastures where animals are not isolated from each other, thereby facilitating gene flow from exotic to indigenous populations. Structured breeding programs can be developed where indigenous rams are shared among farmers in a community in an attempt to curb gene-flow from exotic breeds. Molecular tests can determine relatedness between these indigenous rams to ensure unrelatedness.

### 3.3. Marketing

Adherence by smallholder sheep farmers with the low-input system that is already in place will allow them to market their products as organic or free range, provided that it is of suitable quality. In South Africa, there is already a market for such organic/free range products. Karoo-certified lamb is another niche market that smallholder sheep farmers can supply to, provided that they fall in the demarcated geographical region. It should be feasible to obtain certification, especially in the case of smallholder farmers willing to move from subsistence to emerging/commercial farming practices. Marketing their products on commercial markets will allow the farmers to fetch a higher price for their products and also to supply niche markets. Given their numbers, such farmers could potentially leverage reliable long-term contracts by forming cooperatives, where they pool their stocks and guarantee their clients a sustainable long-term supply. This setup will allow funding agencies to easily provide assistance, as it will not be given to the individual farmers. Weighing and loading facilities can be built on the premises of the cooperative, facilitating effortless marketing of animals by farmers.

### 3.4. Feed Resources

Since indigenous sheep are small-framed, they typically have low maintenance requirements and the ability to survive in harsh environments. In SADC, due to the large areas experiencing hyper-arid, arid, and semi-arid conditions, sheep farming is mostly extensive. Sheep raised under such conditions depend on natural vegetation for their subsistence. Fattening programmes can be strategically developed for those farmers using locally available feedstuffs. Land can be allocated to plant permanent and annual pastures or fodder crops for the animals to supplement natural pasture, while crop residues may be used strategically. Feed formulation can be done by experts to develop low-cost diets based on locally available feedstuffs for fattening. Also, meat, being largely produced from local feed resources, will assist in ensuring that this product is healthy and could putatively be branded and sold at a premium and fetch a higher price in the market. Since all feed resources will be locally produced, they will be low-cost, which will allow farmers to sell at a low price while still making profits.

### 3.5. Health and Diseases

Production in smallholder production systems is constantly under the threat of poor flock health, parasites, and diseases, which could result in a loss of weight, reduced milk production, and poor product quality. This could be due to smallholder farmers introducing more exotic breeds that are not robust to their farming systems since it has been shown that indigenous breeds are more robust and would adapt better to low-input farming systems. Indigenous breeds are also tolerant to seasonal weight loss [34] and have higher poly-unsaturated fatty acids in their meat [31]. Organisation of farmers into cooperatives will facilitate the provision of veterinary services and training by reducing cost and management effort. Farmers should be educated about common diseases and parasites in their production environments and be trained on how to deal efficiently with cases when they occur. Inspection of all animals and a stock inventory should be taken regularly so that clinical disease problems can be handled speedily. One dipping facility can be provided for a single cooperative or a few cooperatives located close together. Proven indigenous knowledge systems on the treatment of certain diseases should be incorporated for sustainable parasite and disease control.

### 3.6. Gender and Age Issues

According to surveys, most smallholder sheep farmers are older males. Females and the youth are not actively involved in sheep farming. Since sheep farming is often practiced in remote areas under marginal conditions and often involves hard work and continuous supervision, it is in many ways a lifestyle occupation. It is perhaps not surprising that young people and females do not find this secluded, responsible and around-the-clock occupation alluring. Sheep farming will need to become more socially interactive, less intense and associated with fewer risks to attract greater numbers of people from these groups. The automated capturing of phenotypic data may provide a platform for farmers to become more involved in the smallholder sheep farming community, while also allowing for extrapolation to social media platforms, should it be desired. The emergence of young entrepreneurs from rural farming communities in agribusiness as leaders can make agriculture more attractive to youth and women.

## 4. Possible Ideas for Innovations

The value of indigenous breeds in terms of adaptability and robustness can be used as a selling point for smallholder farmers. If successful breeding programs are sustained by cooperatives, then semen from indigenous breeds can be sold and/or exported to commercial farmers to improve adaptive traits by incorporation in crossbreeding programs with exotic breeds. This may potentially increase the income value derived from indigenous ovine genetic resources.

Young men and women farmers can start agribusinesses using social media platforms to market and sell sheep products, such as wool, meat, and skins.

## 5. Conclusions

This contribution highlighted a paucity of information on indigenous ovine genetic resources utilized in smallholder farming systems as well as their breeding, husbandry, and management. Fortunately, South Africa has a well-developed and competent commercial sheep industry from where guidelines to deal with these issues could be sourced and adapted. Providing training to youth and women on agriculture and its role in food security should also be prioritised. For this to be possible, investment from the private sector and government is necessary to provide enabling support to smallholder farmers. Fieldworkers should be employed from within the farming communities by private or governmental agencies that are solely dedicated to recording data. It is of paramount importance to develop and extend an affordable and user-friendly recording protocol to allow the accrual of economically important data to base sound selection and management decisions upon. The potential contribution of applicable and informative records from such a system on the further development of a viable smallholder sheep industry cannot be overemphasized. The cultural and economical value of indigenous ovine resources should be highlighted to ensure their conservation, especially ovine breeds that are already at risk of endangerment.

This paper was presented at a colloquium on animal genetic resources in Southern Africa and the sustainable utilization plan that arose from the colloquium are presented in Table 2.

**Table 2.** Sustainable utilisation plan to execute.

| What | How | Who | When |
|---|---|---|---|
| Community-based breeding programme<br>i. Flock 1: created using Namaqua Afrikaners contributed by the communal farmers<br>ii. Flock 2: created using Dorpers from a research flock or stud farmer<br>Two systems will be implemented and run parallel, namely (i) conservation and (ii) terminal crossbreeding. The conservation system will be done to maintain the indigenous breeds, and the terminal will be for marketing the crossbreds with improved weight. | Two nucleus flocks to be established<br><br>• Researchers set up the nucleus flocks, design breeding programmes and provide training<br>• Communal and commercial farmers contribute animals to build the nucleus flocks and manage the breeding programmes<br>• Extension officers will train communal farmers<br>• Government, private sector and non-governmental organisations provide finance for the programmes | • Five years to set up nucleus flocks<br>• Five years for the programme to be self-sustainable | |
| Nutrition for animals | • Rehabilitate natural deteriorated vegetation and pastures<br>• Plant local feed plants, such as indigenous legumes, to provide supplementary feeding<br>• Use indigenous knowledge systems available in communities | • Researchers to conduct veld condition assessment and develop programmes for veld rehabilitation when necessary<br>• Researchers will develop grazing plans for the animals and train farmers<br>• Researchers will identify plants that can be grown for supplementary feeding<br>• Farmers will provide indigenous knowledge on suitable plants for feeding and feeding programmes<br>• Extension officers will provide training to the communal farmers<br>• Government, private sector and non-governmental organisations will finance the feeding programmes | Five years |
| Land availability | Land will be made available for<br>i Setting up and managing nucleus flocks<br>ii. Grazing for animals<br>iii. Growing supplementary feeding<br>iv. Building market facilities | • Local government and traditional leaders will provide land for these activities | Two years |

**Table 2.** *Cont.*

| What | How | Who | When |
|---|---|---|---|
| Health and diseases | • Animals will be dipped and dosed routinely to control external and internal parasites<br>• Daily monitoring of animals for any signs of illness and during mating and lambing seasons will be in place | • Local state veterinary personnel will treat sick animals and dispose of any dead animals<br>• Training will be provided by state veterinary personnel to the communal farmers and extension officers<br>• Communal farmers will monitor the animals every day<br>• Extension officers will monitor the animals on a weekly basis | Five years for farmers to be self-reliant |
| Gender and age issues | • Awareness campaigns on gender and age will be conducted to promote the involvement of women and young people in sheep farming.<br>• This will be done through workshops, advertising, education.<br>• Incorporation of the communal production systems in tertiary education curricula will help extend knowledge and create interest on this production system among the youth<br>• Sheep farming will be made interesting and fun to be involved in through innovation by incorporating modern technology<br>• A Recording and management application (REMAP) will be created for the animals, where all animals in the different stages, e.g., dry ewes, suckling ewes, pregnant ewes, weaners, etc., will be included in the app.<br>• The farmer will be able to view the different categories and animals in those categories on his farm.<br>• Once an animal is born, it will move through the stages within the recording and management application, and then the farmer will be notified when it is due for weaning, mating, marketing, etc.<br>• Also, the farmer will be notified by the recording and management app when it is time to provide supplementary feeding | • Government, Non-governmental organisations (NGOs) and the private sector to fund the awareness campaigns<br>• Incorporation of communal sheep production systems into curricula will be done by tertiary education institutions<br>• Development of REMAP will be a collaboration among researchers, tertiary institutions, farmers, extension officers, government, NGOs, and the private sector. | Five to ten years |

**Table 2.** *Cont.*

| What | How | Who | When |
| --- | --- | --- | --- |
| Market access | • Market sheep from low-input systems as organic/free range against a higher premium, here REMAP will be useful to obtain records of the individuals for traceability<br>• Animals will also be sold on REMAP, where buyers and sellers will meet on this virtual platform. REMAP will have detailed information about an animal, including weights, date of birth and other phenotypic measurements.<br>• Marketing will be improved by building marketing facilities, i.e., auction pens<br>• Providing transport for the animals to the auction facilities<br>• Contracts will also be arranged between abattoirs and farmers for premium animals derived from terminal crossbreeding systems<br>• Farmers will also sell semen to commercial farmers in comparable production environments<br>• Semen will also have its catalogue placed on REMAP, where buyers will be able to access it. | • Researchers and organic certification companies<br>• Government, NGOs and private sector to build marketing facilities, e.g. auction pens<br>• Farmers to pay for their transport costs to the market<br>• Capturing of information onto REMAP will be done by trained personnel | Five years to build market facilities and to ensure contracts for farmers |

**Author Contributions:** The conceptualization and writing the original draft of the review was done by A.H.M.; writing, review, and editing was done by B.D.; further writing and supervision was provided by S.W.P.C. All authors have read and agreed to the published version of the manuscript.

**Funding:** This research forms part of the AU-IBAR project. The APC was funded by Stellenbosch University, Faculty of Agrisciences.

**Acknowledgments:** We would like to acknowledge the guest editor Philip Spongenberg for inviting this review paper. Further on we would like to give our appreciation to Kennedy Dzama for being the instigator of the Animal Genetic Resources colloquium from which this review had its birth.

**Conflicts of Interest:** The authors declare no conflict of interest.

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
