# Peer review of "The Current Status of Indigenous Ovine Genetic Resources in Southern Africa and Future Sustainable Utilisation to Improve Livelihoods"

_diversity, doi:10.3390/d12010014_

Round 1
Reviewer 1 Report
Review Molotsi et al. Ovine genetic resources in Southern Africa
This paper gives an overview of indigenous sheep resources in Southern Africa. It is a nice overview and will be of interest to many of te readers of ‘Diversity’. The only problem I have with the paper is that it is only text without figures and tables (apart from table 1 which is very long and more suited as an appendix). Because of that it is at some points a bit hard to follow, especially for readers not familiar with the situation in Southern Africa. For example, around L. 60-65, a simple map with the different climatic regions and location of (some of) the breeds mentioned would help, and will make at the same time the paper more inviting to read. An illustration of the three types of sheep distinguished (L. 85-86) would also help.
The paper deals with a problem that probably is faced by all livestock resources across the world, how to conserve and value indigenous resources in the presence of exotic high input/high output breeds. Authors may refer to Gizaw et al. (2008), Genet. Sel. Evol. 40 (2008) 433–447, where Ethiopian sheep breeds are evaluated taking not only production aspects into account, but also other important aspects such as ecology and farmers appreciation. A similar approach for South African sheep would be valuable.
However, overall the paper is interesting and a welcome contribution to the journal, so I suggest acceptance after some modification.
Author Response
Dear Reviewer
Thank you for taking the time out to review our paper. We have considered your recommendations and have added a map as Figure 1 in text to describe the SADC regions and climatic conditions.
The paper of Gizaw et al., 2008 has also been considered and we have concluded that a similar approach to conservation of indigenous breeds in SA would be valuable.
Reviewer report 1
This paper gives an overview of indigenous sheep resources in Southern Africa. It is a nice overview and will be of interest to many of the readers of ‘Diversity’. The only problem I have with the paper is that it is only text without figures and tables (apart from table 1 which is very long and more suited as an appendix). Because of that it is at some points a bit hard to follow, especially for readers not familiar with the situation in Southern Africa. For example, around L. 60-65, a simple map with the different climatic regions and location of (some of) the breeds mentioned would help, and will make at the same time the paper more inviting to read. An illustration of the three types of sheep distinguished (L. 85-86) would also help.
Author response: This has been addressed We have included the list of SADC countries in line 31-34 as well as a Map on the different climatic regions and photo’s of breeds in these regions. See Figure 1 L91-93
The paper deals with a problem that probably is faced by all livestock resources across the world, how to conserve and value indigenous resources in the presence of exotic high input/high output breeds. Authors may refer to Gizaw et al. (2008), Genet. Sel. Evol. 40 (2008) 433–447, where Ethiopian sheep breeds are evaluated taking not only production aspects into account, but also other important aspects such as ecology and farmers appreciation. A similar approach for South African sheep would be valuable.
Author response: See L 218-225
However, overall the paper is interesting and a welcome contribution to the journal, so I suggest acceptance after some modification.
Reviewer 2 Report
Dear Editor,
Thank you for the invitation to review this manuscript.
The topic is indeed interesting and addresses the ovine genetic resources of SADC countries. The authors focus on major traits and have very interesting considerations on the use of these breeds, their value and their preservation.
I believe the mansucript should be accepted for publication, providing the following aspects are met:
1) Add a map of SADC countries and a distribution of the major indigenous genetic resources
2) Please provide a figure with pictures of some of the most important and unknown genetic resources, particularly those that are mentioned throughout the text.
3) Section 2.2 must be improved with data on seasonal weight loss tolerance physiology and functional genomics studies that are available for the Damara breed:
https://www.ncbi.nlm.nih.gov/pubmed/23031388
https://www.ncbi.nlm.nih.gov/pubmed/23345065
https://www.ncbi.nlm.nih.gov/pubmed/24204803
https://www.ncbi.nlm.nih.gov/pubmed/26037039
https://www.ncbi.nlm.nih.gov/pubmed/26828937
https://www.ncbi.nlm.nih.gov/pubmed/27966615
https://www.ncbi.nlm.nih.gov/pubmed/28366878
https://www.ncbi.nlm.nih.gov/pubmed/30785939
https://www.ncbi.nlm.nih.gov/pubmed/31250490
https://www.sciencedirect.com/science/article/pii/S0921448812003574
4) The authors seem to have ignored interesting information on performance and traits of Damara and other SADC sheep breeds. They should include the following references in the text:
https://www.ncbi.nlm.nih.gov/pubmed/3786655
https://www.ncbi.nlm.nih.gov/pubmed/31380447
https://www.ncbi.nlm.nih.gov/pubmed/27038194
https://www.ncbi.nlm.nih.gov/pubmed/26767563
https://www.ncbi.nlm.nih.gov/pubmed/21725705
https://www.ncbi.nlm.nih.gov/pubmed/21544704
https://www.ncbi.nlm.nih.gov/pubmed/22063250
https://www.ncbi.nlm.nih.gov/pubmed/26178370
https://www.sciencedirect.com/science/article/pii/S0921448899000760
5) The authors have also ignored the Dorper. This is the most important sheep breed arising from Southern Africa. The only one with an international distribution. I think this should be mentioned in the text and some reviews should be cited:
https://www.sciencedirect.com/science/article/pii/S0921448899001546
https://www.sciencedirect.com/science/article/pii/S0921448899001558
https://www.sciencedirect.com/science/article/pii/S092144889900156X
https://www.sciencedirect.com/science/article/pii/S0921448899001571
https://www.sciencedirect.com/science/article/pii/S0921448899001583
https://www.sciencedirect.com/science/article/pii/S0921448899001595
6) Table 1 is very difficult to follow. Turn it into figure(s) highlighting the major aspects. Information in table 1 should be addressed in a separate section in the body of the texto.
7) Remove the last paragraph of the conclusions. It makes no sense relating it to a conference presentation.
Author Response
Dear reviewer
Reviewer report 2
1) Add a map of SADC countries and a distribution of the major indigenous genetic resources
Author response: A map has been added, See Figure 1, L 91-93
2) Please provide a figure with pictures of some of the most important and unknown genetic resources, particularly those that are mentioned throughout the text.
Author response: Pictures of fat-tailed, thin-tailed and fat-rumped sheep have been added to Figure 1, L 91-93.
3) Section 2.2 must be improved with data on seasonal weight loss tolerance physiology and functional genomics studies that are available for the Damara breed:
Author response: We have included some of the literature below on the attributes of the Damara breed. See L151-164
4) The authors seem to have ignored interesting information on performance and traits of Damara and other SADC sheep breeds. They should include the following references in the text:
Author response: We have included the information on the reproductive performance of the Damara since this was consistent with the traits discussed for other indigenous breeds.
5) The authors have also ignored the Dorper. This is the most important sheep breed arising from Southern Africa. The only one with an international distribution. I think this should be mentioned in the text and some reviews should be cited:
Author response: Since the Dorper is a composite breed developed in SA, we did not focus on it. Many studies have been done on the production performance of this breed. Literature sources cited have, however, compared most of the breeds in the text to the Dorper.
5) Table 1 is very difficult to follow. Turn it into figure(s) highlighting the major aspects. Information in table 1 should be addressed in a separate section in the body of the texto.
7) Remove the last paragraph of the conclusions. It makes no sense relating it to a conference presentation.
Author response: This paper forms part of five papers that have been submitted to this special issue of Rare livestock breeds. The Sustainable utilization plan arose from the colloquium mentioned and each of the five papers was required to propose a sustainable utilization plan.
Thank you so much for taking the time out to review our paper. Your inputs are highly appreciated. We have tried to include most of the literature that was recommended on the Damara and other indigenous breeds. Most of the literature given was on functional genomics and metabolomics which was not the main focus of the paper. Since the focus was mainly on genetic diversity of the indigenous breeds in SADC regions as well as their phenotypic characteristics.
The Dorper in SA is seen as a commercial breed, even though it is indigenous to SA and transboundary. Therefore, it is mentioned in the paper but more in comparison to other indigenous breeds that are not widely used in our production systems.
Reviewer 3 Report
In the article “The current status of indigenous ovine genetic resources in Southern Africa and future sustainable utilisation to improve livelihoods”, authors make an appraisal on the situation of sheep breeds in southern Africa region, based on a literature review. The document is relatively well written, however I see at least several important points missing on it.
First, it does not provide in a clear way a real appraisal of what are those breeds raised in the different countries. There is no list of breeds (in DAD-IS data base, there seems to be around 40 local and regional sheep breeds), not even a list of countries, which is a problem as most of the readers are not necessarily aware of what are the countries in Southern Africa. Secondly, authors miss the point that a major point for the maintenance of local breeds relate with keeping demographic numbers. Working on genetic variability and effective population size should come after. In that extent, a big challenge is that information on population size is unknown for many breeds, even if the situation is largely contrasted across southern Africa countries. This definitively deserves more discussion. There are also very limited information of livestock keepers of those breeds, in term of numbers and production systems. Are those breeds raised in backyard, mixed systems, pastoralist systems?
Various remarks
L65 It is bit irrelevant to talk about effective population size if already nothing is known about demographic population size.
L87 please use heterozygosity instead of Hz
L99 For most of the people “PCA Analysis at K=9” will be meaningless. Please remove.
L160 before even talking about breeding programs and conservation, a first prerequisite is to be able to characterize and monitor local populations. It is rather unclear whether the different breeds are monitored by countries in the region.
L191-200 Before any statement on Quality Signs, authors should state if (i) there is a market for quality products (are consumers interested for such products in the countries of the region)? Is there a legal framework for certification in the countries?
L201-210 This depends on the production system where sheep are raised, which have not been precised.
L211-221 Authors should make clear in what extent this is relevant for local breeds.
Table 1. Bullets are not consistent across the table. Please remove a space between “Health and Diseases”
Author Response
Dear Reviewer
Reviewer report 3
Reviewer comment: First, it does not provide in a clear way a real appraisal of what are those breeds raised in the different countries. There is no list of breeds (in DAD-IS data base, there seems to be around 40 local and regional sheep breeds), not even a list of countries, which is a problem as most of the readers are not necessarily aware of what are the countries in Southern Africa.Author response: We have included the list of SADC countries in line 31-33. There is no list of breeds as the author did refer to the 109 breeds as listed in the AU-IBAR reference in line 35-36. However, we have included a Table with the list of only indigenous breeds to SADC. The breeds list on the DAD-IS database includes both exotic and indigenous sheep breeds. Since the focus of the paper is mainly on indigenous breeds (See Table 1 Line 95) they were emphasised.
Reviewer comment: Secondly, authors miss the point that a major point for the maintenance of local breeds relate with keeping demographic numbers. Working on genetic variability and effective population size should come after. In that extent, a big challenge is that information on population size is unknown for many breeds, even if the situation is largely contrasted across southern Africa countries. This definitively deserves more discussion.
Author response: We included Table 1 with suggested population sizes for the different breeds and discussed this issue. See Line 70-74
Reviewer comment: There are also very limited information of livestock keepers of those breeds, in term of numbers and production systems. Are those breeds raised in backyard, mixed systems, pastoralist systems?
Author response: These different farming systems have been widely reviewed and AU-IBAR published a book discussing the different farming systems in detail. Also see the review of Molotsi et al. (2017) in Sustainability. Since the paper is focused more on the genetic diversity and diversity of ovine breeds, we have chosen not to go into the detail of the farming system. However, it was mentioned in Line 251-253 that the majority of sheep farmers in SADC are occupied in extensive communal farming systems.
L65 It is bit irrelevant to talk about effective population size if already nothing is known about demographic population size.
Author response: See line 70-74
L87 please use heterozygosity instead of Hz
Author response: Hz was replaced with heterozygosity L105
L99 For most of the people “PCA Analysis at K=9” will be meaningless. Please remove.
Author response: PCA Analysis at K=9 were removed L117
L160 before even talking about breeding programs and conservation, a first prerequisite is to be able to characterize and monitor local populations. It is rather unclear whether the different breeds are monitored by countries in the region.
Author response: See L196-201
L191-200 Before any statement on Quality Signs, authors should state if (i) there is a market for quality products (are consumers interested for such products in the countries of the region)? Is there a legal framework for certification in the countries?
Author response: See L238-240.
L201-210 This depends on the production system where sheep are raised, which have not been precised.
Author response: Majority of the sheep are raised in an extensive production systems, where sheep graze on natural vegetation. See L252-253.
L211-221 Authors should make clear in what extent this is relevant for local breeds.
Author response: This is relevant to a lesser extent for local breeds, since they are known to show disease resistance. However, in terms of product quality their carcass yield might be undesirable, but the nutritional quality might be better in terms of fatty acid composition to exotic breeds. L265-267.
Table 1. Bullets are not consistent across the table. Please remove a space between “Health and Diseases”
Author response: The space between health and diseases has been removed. See L312. The table (now Table 2) was reformatted to be aligned left and lines were inserted between the sections to allow easier interpretation.
We want to thank you for taking the time out to review this paper. Your inputs are highly appreciated.
Round 2
Reviewer 3 Report
In this new version of the article “The current status of indigenous ovine genetic resources in Southern Africa and future sustainable utilisation to improve livelihoods”, authors made effort to provide more information on their breeds, which I appreciate.
I think some clarification are however needed before acceptance.
In particular:
Table 1 is too large.
Figure 1 requires improvement. One figure is for instance without legend.
L73-74 Sentence unclear. Does this refer to the effective population size of Southern African breeds, and if yes which ones.
Author Response
Comments and Suggestions for Authors
In this new version of the article “The current status of indigenous ovine genetic resources in Southern Africa and future sustainable utilisation to improve livelihoods”, authors made effort to provide more information on their breeds, which I appreciate.
I think some clarification are however needed before acceptance.
In particular:
Table 1 is too large.
I would have it subdivided between either Exotic, Indigenous and Synthetic or according to sheep type. The way demographic numbers are presented would require improvements.
Author response: I have removed all the exotic breeds from the table, since the focus is mainly on indigenous ovine genetic resources to reduce the Table size. I think it will make it difficult to order the table via exotic, indigenous and synthetic, it will take more space up to allocate to the different countries and population numbers.
For instance, subdividing between two columns, Min and Max.
Author response: I divided min and max into different columns, this helped to reduce the size of the Table.
I would also suggest either to indicate a data for the population estimate, or to precise a threshold, (for instance all estimate made over the last ten years).
Author response: I have included a date for when the numbers have been estimated. See Table 1 and line 76-78.
The exotic/Indigenous/Synthetic column require to be harmonized: What is the difference between a developed breed, a composite breed, a developed in SA and a synthetic one?
Author response: A synthetic breed is similar for composite breed, it refers to breeds that have been created through the crossing of different breeds. I have chosen to use the term synthetic throughout the text and Table when referring to these breeds.
Developed breeds refer to exotic breeds that have been imported and further selected by breeders in SADC to adapt to the specific environment where they are found. See line 69-73
Besides the fact that a majority of breeds are of unknown status, authors should comment whether the ones with known status are endangered or not.
Author response: The status of those that are not at risk is made clear in the Table, this is an indication that they are not endangered
Figure 1 requires improvement. One figure is for instance without legend.
Author response: I have provided an abbreviation list with all the SADC countries’ names.
L73-74 Sentence unclear. Does this refer to the effective population size of Southern African breeds, and if yes which ones.
Author response: I have included the breeds that I am referring too, see Line 78-82